# Umbilical Cord Mesenchymal Stromal Cells for Steroid-Refractory Acute Graft-versus-Host Disease

**DOI:** 10.3390/ph16040512

**Published:** 2023-03-30

**Authors:** Camila Derminio Donadel, Bruno Garcia Pires, Nathália Cristine André, Thalita Cristina Mello Costa, Maristela Delgado Orellana, Sâmia Rigotto Caruso, Adriana Seber, Valéria Cortez Ginani, Alessandra Araújo Gomes, Yana Novis, George Maurício Navarro Barros, Neysimélia Costa Vilella, Gláucia Helena Martinho, Ana Karine Vieira, Andrea Tiemi Kondo, Nelson Hamerschlak, Jayr Schmidt Filho, Erick Menezes Xavier, Juliana Folloni Fernandes, Vanderson Rocha, Dimas Tadeu Covas, Rodrigo Tocantins Calado, Renato Luiz Guerino-Cunha, Gil Cunha De Santis

**Affiliations:** 1Regional Blood Center of Ribeirão Preto, Ribeirão Preto Medical School, University of São Paulo, Ribeirão Preto 05508-220, Brazil; camila.donadel@hemocentro.fmrp.usp.br (C.D.D.); nathalia.andre@hemocentro.fmrp.usp.br (N.C.A.); mdelgado@hemocentro.fmrp.usp.br (M.D.O.); samia@hemocentro.fmrp.usp.br (S.R.C.); dimas@hcrp.usp.br (D.T.C.); rtcalado@usp.br (R.T.C.); 2Department of Medical Imaging, Hematology and Oncology, Ribeirão Preto Medical School, University of São Paulo, Ribeirão Preto 05508-220, Brazil; bgppsilva@hcrp.usp.br (B.G.P.); tcmcosta@hcrp.usp.br (T.C.M.C.); rlgc@usp.br (R.L.G.-C.); 3Grupo de Apoio ao Adolescente e à Criança com Câncer (GRAACC), Universidade Federal de São Paulo (UNIFESP), São Paulo 04021-001, Brazil; adrianaseber@gmail.com (A.S.); valeriaginani@gmail.com (V.C.G.); 4Hospital Samaritano, São Paulo 01232-010, Brazil; 5Hospital Sírio Libanês, São Paulo 01308-050, Brazil; ale_a_gomes@hotmail.com (A.A.G.); yananovis@yahoo.com.br (Y.N.); 6Barretos Cancer Hospital, Barretos 14784-400, Brazil; georgenavarrobr@yahoo.com.br (G.M.N.B.); ncvilella@hotmail.com (N.C.V.); 7Hospital das Clínicas da Universidade Federal de Minas Gerais, Belo Horizonte 30130-100, Brazil; ghelenamar@gmail.com (G.H.M.); ana.karine@ebserh.gov.br (A.K.V.); 8Hospital Israelita Albert Einstein, São Paulo 05652-900, Brazil; andrea.kondo@einstein.br (A.T.K.); hamer@einstein.br (N.H.); julianafolloni@gmail.com (J.F.F.); 9Hospital A.C.Camargo Cancer Center, São Paulo 01525-001, Brazil; jayr.filho@accamargo.org.br; 10Hospital das Clínicas da Faculdade de Medicina da Universidade de São Paulo (FMUSP), São Paulo 05403-010, Brazil; erick.xavier@hc.fm.usp.br (E.M.X.); vanderson.rocha@hc.fm.usp.br (V.R.); 11Instituto da Criança do Hospital das Clínicas da Faculdade de Medicina da Universidade de São Paulo, São Paulo 01246-903, Brazil

**Keywords:** mesenchymal stromal cell, hematopoietic stem cell transplantation, acute graft-versus-host disease

## Abstract

Background: Steroid-refractory acute graft-vs.-host disease (SR-aGVHD) is a complication of allogeneic hematopoietic stem cell transplantation with a dismal prognosis and for which there is no consensus-based second-line therapy. Ruxolitinib is not easily accessible in many countries. A possible therapy is the administration of mesenchymal stromal cells (MSCs). Methods: In this retrospective study, 52 patients with severe SR-aGVHD were treated with MSCs from umbilical cord (UC-MSCs) in nine institutions. Results: The median (range) age was 12.5 (0.3–65) years and the mean ± SD dose (×10^6^/kg) was 4.73 ± 1.3 per infusion (median of four infusions). Overall (OR) and complete response (CR) rates on day 28 were 63.5% and 36.6%, respectively. Children (*n* = 35) had better OR (71.5% vs. 47.1%, *p* = 0.12), CR (48.6% vs. 11.8%, *p* = 0.03), overall survival (*p* = 0.0006), and relapse-free survival (*p* = 0.0014) than adults (*n* = 17). Acute adverse events (all of them mild or moderate) were detected in 32.7% of patients, with no significant difference in children and adult groups (*p* = 1.0). Conclusions: UC-MSCs are a feasible alternative therapy for SR-aGVHD, especially in children. The safety profile is favorable.

## 1. Introduction

Allogeneic hematopoietic stem cell transplantation (alloHSCT) is a potential curative treatment option for many hematological diseases (Snowden et al., 2022). Nevertheless, it is commonly associated with complications such as infection, immunosuppression, and acute graft-versus-host disease (aGVHD) [1,2,3].

aGVHD affects approximately 40% of patients undergoing alloHSCT, varying according to the type of donor, cell source, and method of prophylaxis [4,5,6,7,8,9]. aGVHD is the leading cause of short-term death in patients undergoing alloHSCT and is characterized by acute inflammation and organ dysfunction [10]. The organs most frequently affected are the skin (maculopapular rash), the gastrointestinal tract (nausea, vomiting, anorexia, watery or bloody diarrhea, and abdominal cramps), and the liver (cholestasis, manifested by hyperbilirubinemia), typically occurring in the first 100 days of transplantation, but occasionally starting after this period, even superimposed to chronic GVHD, the so-called overlap syndrome [11]. The diagnosis of aGVHD is based on clinical and anatomopathological findings of the affected tissue or organ [12]. The most used classification for aGVHD severity is the Mount Sinai Acute GVHD International Consortium (MAGIC) [13]. Each affected organ (skin, liver, upper gastrointestinal tract, and lower gastrointestinal tract) is individually graded based on the severity of involvement, ranging from stage 1 (mild) to stage 4 (very severe), and the disease is graded globally from grade I (mild) to IV (very severe). Grades III-IV confer, as expected, a higher mortality rate [10].

Corticosteroids (prednisone or methylprednisolone at a dose of 1–2 mg/kg/day) are the first-line therapy for this complication; however, 40–50% of patients are refractory to this class of drug [14,15]. Steroid-refractory aGVHD (SR-aGVHD) is associated with a dismal prognosis, with a survival rate, considering all grades, of 40% at six months [14], and as low as 17% for long-term survival for non-responders to therapy [16]. Recently, the inhibitor of JAK2 ruxolitinib was approved by the Food and Drug Administration (FDA) for the second-line treatment of aGVHD, as it results in a response rate in approximately two-thirds of patients [17,18]. Several other agents were evaluated; however, none provided an entirely satisfactory clinical response [19]. An interesting alternative as a second-line treatment for SR-aGVHD could be the administration of mesenchymal stromal cells (MSCs).

MSCs are non-hematopoietic cells of mesodermal origin that are present in most body tissues, including bone marrow, adipose tissue, umbilical cord, and placenta, from which they can be extracted and expanded ex vivo [20,21]. MSCs can basically be used for two purposes: to regenerate tissues and organs [22,23] and for immunomodulation in immune-mediated disorders, such as autoimmune diseases and aGVHD [24,25]. MSCs diminish inflammation intensity and induce immune suppression by, at least in part, suppression of the proliferation of T and NK cells and reduction of the number of pro-inflammatory monocytes. Additionally, MSCs promote regulatory T cell expansion [26]. In 2004, Le Blanc et al. reported a successful case of MSC use for aGVHD [27]. Later, the same group described a series of cases in which MSC administration for aGVHD provided excellent disease control [28]. Subsequently, Kurtzberg et al., in two important studies, showed benefits of MSC administration for children with aGVHD. The first study was a one-arm phase 3 clinical trial that showed encouraging (overall response in 70.4% of the patients at day 28) clinical benefits for patients treated with MSCs [29]. The second study, an expanded access program, included 241 patients with an overall response rate at day 28 and overall survival at day 100 of 63.1% and 66.9%, respectively, a result much better than those previously reported in the literature with other types of therapy [30]. Other studies also showed encouraging results, which contributed to the approval of MSC product use in children with SR-aGVHD in Canada, New Zealand, and Japan [31]. Because of some encouraging results obtained so far, the European Society for Blood and Marrow Transplantation recommended MSC administration as second-line therapy (category 2A) for acute GVHD [32]. A retrospective study of bone marrow MSC use in SR-aGVHD developed by our group showed a 2-year survival rate of 17%, in which all survivors had a complete response [33]. These studies employed bone marrow as the source of MSC. Nevertheless, the use of the umbilical cord as a source seems to be associated with similar results [34].

MSC administration has been associated with a few adverse events, usually of mild presentation [29,30,33]. Besides the favorable safety profile, MSC products can be manufactured relatively easily in centers that have a laboratory dedicated to cellular therapy. Additionally, MSC products can be manufactured in countries in which other biologicals for second-line therapy for this disease are not readily available or are totally unavailable.

In this study, we retrospectively analyzed the results of umbilical cord-derived MSC administration as a salvage therapy for patients with SR-aGvHD.

## 2. Results

### 2.1. Patients’ Characteristics, Infused Cell Dose, and Schedule

A total of 52 patients with SR-aGVHD were included in the analysis according to the inclusion and exclusion criteria. Two patients were excluded from the analysis, one because of a lack of data and another one because he died of sepsis within 24 h of MSC infusion (Figure 1). The median (range) age at transplantation was 12.5 years (0.3–65), and males made up the majority (65.4%) of the study group. The median (range) follow-up was 4.6 (0–92) months. The median (range) time from diagnosis to MSC infusion, and the number of infusions were 36.5 (4–250) days and 4 (1–11), respectively. The mean ± SD dose per infusion was 4.73 × 10^6^/kg ± 1.30 × 10^6^/kg. Time to infusion, number of infusions, and MSC dose had no statistical difference between children and adults.

Table 1 summarizes the patient characteristics of the overall population, infused cell dose, and schedule, according to age, <18 years (children) and ≥18 years (adults).

### 2.2. Efficacy

#### 2.2.1. Overall Response (OR)

A total of 63.5% of the patients (children, 71.5%; adults, 47.1%) had a favorable response on day 28 after MSC infusion, which progressively declined throughout the period of evaluation (2 years) until reaching only 26.9% (children 34.3%; adults 11.8%) (Table 2).

#### 2.2.2. Overall (OS) and Relapse-Free Survival (RFS)

Children presented a much better OS and RFS than adults (*p* = 0.0006 and *p* = 0.0014, respectively) (Figure 2). Moreover, the patients who had a favorable response on day 28 after MSC infusion had also a better OS (*p* < 0.0001) (Figure 3). Total OS at days 100 and 180 were 51.9% (27/54) and 38.5% (20/54), respectively. On day 100, OS in children and adults was 62.9% (22/35) and 29.5% (5/17), respectively (*p* = 0.038). On day 180, OS in children and adults was 48.6% (17/35) and 17.6% (3/17), respectively (*p* = 0.038). Finally, considering only patients who died (35/52; 67,3%), we observed that children and adults took 68.5 (12–547) days and 25.0 days (2–644) from the first MSC infusion to death, respectively (*p* = 0.02).

#### 2.2.3. Chronic GVHD

The occurrence of chronic GVHD was evaluated in 44 of 52 patients (only those who survived beyond day 100 from HSCT), of which 7/34 (20.6%) children and 3/10 (30%) adults had this complication (*p* = 0.67).

### 2.3. Safety

A total of 17 (32.7%) patients (12 children and 5 adults) presented acute AE associated with MSC infusion, of which only 2 were moderate/severe (1 angioedema and 1 seizure) (according to Common Terminology Criteria for Adverse Events-CTCAE) [35]. There was no difference between the two groups regarding this issue (Table 3). Additionally, cardiovascular acute AE (tachycardia/bradycardia and hypertension/hypotension) was observed in 11.4% of the children and in 54.5% of the adults (*p* = 0.06).

## 3. Materials and Methods

### 3.1. Study Design

This is a multicenter (Hospital das Clínicas da Faculdade de Medicina de Ribeirão Preto—USP, Hospital das Clínicas da Faculdade de Medicina da Universidade de São Paulo—USP, Grupo de Apoio ao Adolescente e à Criança com Câncer—GRAACC, Hospital Samaritano, Hospital Sírio-Libanês, Hospital Israelita Albert Einstein, Fundação Pio XII—Barretos Cancer Hospital, Hospital A.C.Camargo Cancer Center, Hospital das Clínicas da Universidade Federal de Minas Gerais), retrospective study of the efficacy and safety of MSC administration for treatment of SR-aGVHD. The data were collected from the patients’ medical charts.

This study was approved by the institutional review boards of the participating centers and was conducted in accordance with the Declaration of Helsinki.

### 3.2. Patients

Patients with SR-aGVHD grade ≥ II after alloHSCT treated with MSCs between 2015 and 2022 were included in this study. Disease severity was classified by the MAGIC score [13]. SR-aGVHD was defined according to one of the following criteria: disease progression after three days of treatment (methylprednisolone 2 mg/kg/day or equivalent); failure to respond after seven days of treatment; involvement of a new site; or recurrence of GVHD during the corticosteroid tapering [19]. Death during the first 24 h of the MSC infusion and lack of clinical information in the patients’ medical charts were considered exclusion criteria.

### 3.3. Mesenchymal Stromal Cells

MSCs were isolated from the umbilical cord of unrelated healthy neonates, expanded ex vivo and cryopreserved with dimethyl sulfoxide between the third and sixth passage. Each MSC dose was stored in the vapor phase of liquid nitrogen, thawed, and reconstituted in a human serum albumin, saline, and anticoagulant citrate dextrose solution prior to administration, and injected intravenously by central catheter. MSCs must meet the following criteria for clinical use: viability ≥ 70% (it was ≥95% at freezing and ≥80% after thawing); typical immunophenotyping with positive CD73, CD90, CD105, and CD166 markers and absence of expression of hematopoietic and endothelial lineage markers D45, CD34, CD14, CD31, and HLA-DR; and no virus, fungus, bacteria, or mycoplasma contamination.

### 3.4. Variables of Interest

We evaluated the patients’ transplant and MSC infusion characteristics, such as age, sex, diagnosis, conditioning regimen, graft source, days of MSC infusion after a-GVHD diagnosis, number of infusions, and dose of MSCs. Conditioning regimens were classified as myeloablative (MAC) or reduced-intensity (RIC) according to the Center for International Blood and Marrow Transplant Research (CIBMTR) [36]. The severity of aGVHD and the organs affected were also evaluated.

### 3.5. Endpoints

The primary endpoint in this study was the overall response rates, including partial and complete responses, defined according to what was previously described [29]. Response rates were evaluated at days 28, 100, and 180, and at 1 and 2 years after MSC infusion. Secondary endpoints were overall survival, relapse-free survival, frequency of chronic GVHD, and incidence and severity of acute (≤24 h) and adverse events (AE) related to MSC infusion. Another secondary endpoint was the comparison of response rates and survival rates in the pediatric population and in the adult population.

### 3.6. Statistical Analysis

Results were expressed as mean ± standard deviation or median (range) according to distribution characteristics and proportions. When two groups were compared, a two-sided unpaired Student’s *t*-test (for parametric data) or a Mann–Whitney test (for non-parametric data) was used. Categorical variables were presented as frequency (%) and compared using a chi-square test or Fisher’s exact test. Overall survival (OS) estimates were calculated using the Kaplan–Meier method, and differences between the groups were assessed by the log-rank test. Time to events was calculated in days and months from the diagnosis of aGVHD to the day of the first MSC administration, and from this to death or the last date of follow-up, respectively. The results were considered to be statistically different when the *p*-value was below 0.05 (by two-tailed testing). Statistical analyses were performed using statistical GraphPAD Prism software, version 9.5.0.

## 4. Discussion

This non-randomized retrospective observational study showed that the administration of mesenchymal stromal cells obtained from umbilical cord results in substantial clinical benefits in steroid-refractory acute graft-versus-host disease, mainly in children. Approximately two-thirds of our patients presented a 28-day overall response to this therapy, which was in agreement with what was reported in previous studies [29,30,34,37,38] (Table 4). Moreover, the response rate was much higher in children than in adults, similar to that observed by others [37], and similar to the results of the study by Kurtzberg et al., in which the authors evaluated only children [29,30]. Most interestingly, almost half the children presented a 28-day complete response, a higher rate than that observed evaluated by Kurtzberg et al. [29]. A possible explanation for this finding could be the higher MSC dose (the double) than most of the studies, despite the late cell infusion (36.5 days after aGVHD diagnosis). It is possible that a higher cell dose offset the late therapy, the lower number of infusions, and the more severe aGVHD in our patients than the others, as depicted in Table 4. Finally, comparing MSC therapy with ruxolitinib, regarding response rates in children, we observed a similar OR/CR at day 28 in this study to Locatelli et al. (71.5/48.2% vs. 84.4/48.9%, respectively) [39]. Additionally, in the studies REACH1 and REACH2, which evaluated this drug, OR at day 28 was 54.9% and 62%, respectively, similar to our results [17,18].

In our study, OS at day 100 was lower than that observed in the two studies of Kurtzberg et al. [29,30]. Nevertheless, considering only the children of our study, the difference appears to be not significant (62.9% vs. 74.1% and 66.9%). Additionally, OS in children is much better than that observed in adults, which is in accordance with Kebriaei et al. [37]. Furthermore, OS at day 180 was inferior to that observed by Kurtzberg et al. [29] and similar to that found by others [37,38]. Nevertheless, all patients from the study by Kurtzberg et al. received MSC as second-line therapy and earlier than our cohort (12 vs. 36 days), which could be the explanation for the better outcomes [29]. This suggests that MSC infusion should be instituted as soon as possible. Additionally, as expected, OS in the responder group, defined as those who achieved a 28-day OR, was superior to that observed in the non-responder group, as had been previously shown by others [29,38].

It is possible that the administration of MSCs to SR-aGVHD in our patients could have resulted in a slight reduction in chronic GVHD, which affects approximately 50% of patients, a phenomenon that has already been suggested by others [40,41]. In this study, there was no difference between the incidence of chronic GVHD in children and adults.

To indicate MSC use for SR-aGVHD, one needs to take into consideration other agents with known efficacy and acceptable safety profiles. The most important one is the drug ruxolitinib, which was recently approved by the FDA for patients over 12 years of age with this condition; however, complete response rates are still not thoroughly satisfying [17,18]. Our results suggest that MSCs propitiate a similar response rate to that shown with this drug, and, perhaps, an even higher complete response rate considering all the patients herein described. Another important issue is the incidence rate of adverse events, which seems to be higher and more severe with ruxolitinib than with MSC.

Approximately one-third of our patients presented acute AE, mostly mild; however, this was a higher rate than that observed by others [29,34,42,43]. A possible explanation for this finding could be the fact that our patients had more advanced disease and received a higher dose of MSC than that reported in most studies, and later during the evolution of the disease. Despite this, we consider it important to emphasize that the occurrence of severe AE was very low (observed in only two patients in our cohort). An interesting finding was that the adult group appears to be more vulnerable to cardiovascular AE. Additionally, the AE rate in our patients was lower than that observed with the use of ruxolitinib, despite the fact that we did not follow the patients for the occurrence of late-beginning AE [17,18]. MSC use has not been associated with hematological alterations, whereas cytopenia, especially thrombocytopenia, is a common complication of ruxolitinib use. Locatelli et al. observed that approximately half the patients (children) had dose changes or interruptions due to adverse events, which appear to be higher than those found in this study [39].

Another important aspect is the relatively limited access to other drugs for SR-aGVHD in the public health system in our country, such as, for example, the aforementioned ruxolitinib. Moreover, MSC products are administered for a short time, whereas ruxolitinib is usually used for longer periods. It is important to emphasize that ruxolitinib is not yet approved for children under 12 years of age, exactly the age group for which MSC administration was proved to be most efficacious.

MSCs obtained from the umbilical cord are a relatively new source of MSCs, and are considered to be associated with a number of advantages, such as higher MSC proliferative potential, lower risk of transmissibility of infection agents, higher production of units ready for use (off the shelf), and, presumably, a lower risk for the donor [44]. Other interesting sources seem to be pooled bone marrow and decidua MSCs, which are also used to treat aGVHD in children [45,46]. Decidua MSCs were used in only six patients, with durable complete responses seen in four of them [46]. The main disadvantage is that the separation of MSCs from the umbilical cord is more labor-intensive than using bone marrow as the source.

This study suffers from three main limitations, the first one being inherent to retrospective analyses. The other limitations are the relatively small sample size, especially for the adult group and after day 180 from MSC infusion, and the heterogeneity of the patient population, such as diagnosis, and patient care, such as graft source, conditioning regimens, prophylaxis, and second-line treatment for GVHD. Despite these limitations, we had a sample size large enough to show the difference between children and adults regarding the efficacy of MSC for aGVHD and the importance of the response evaluation at day 28.

In conclusion, we believe that umbilical cord mesenchymal stromal cells appear to be a valuable therapeutic modality for steroid-refractory acute graft-versus-host disease, especially for countries in which other agents are less available. Children present a better response rate than adults. Moreover, clinical response on day 28 appears to be fundamental to defining prognosis. Additionally, UC-MSC administration seems to be safe. Our group believes that MSC therapy should be considered for second- or third-line therapy for SR-aGVHD in children. Finally, it would be interesting to investigate the association between the use of MSCs and ruxolitinib, mainly in adults, even as a bridge therapy until the ruxolitinib could be expected to provide beneficial effects.

## Figures and Tables

**Figure 1 pharmaceuticals-16-00512-f001:**
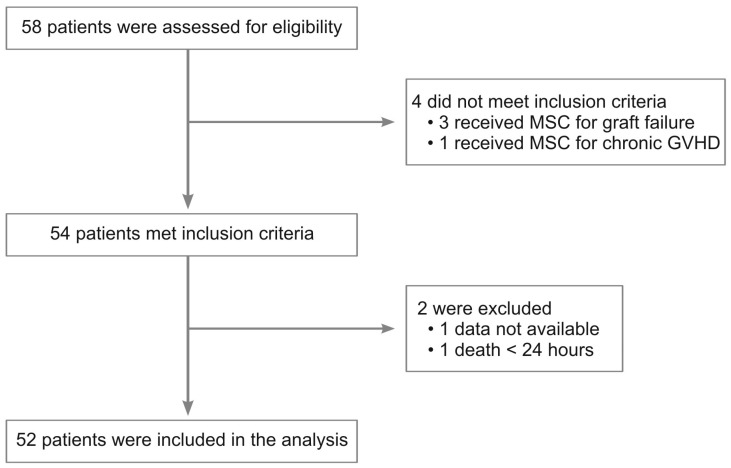
Flowchart showing the reasons for the exclusion of 6 patients, which resulted in a total of 52 patients included in the analyses.

**Figure 2 pharmaceuticals-16-00512-f002:**
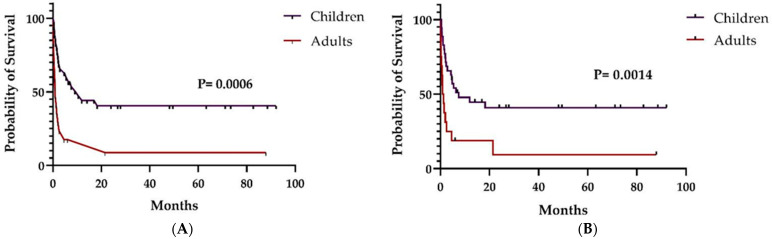
Overall (**A**) and relapse-free survival (**B**) in children and adults (after MSC infusion).

**Figure 3 pharmaceuticals-16-00512-f003:**
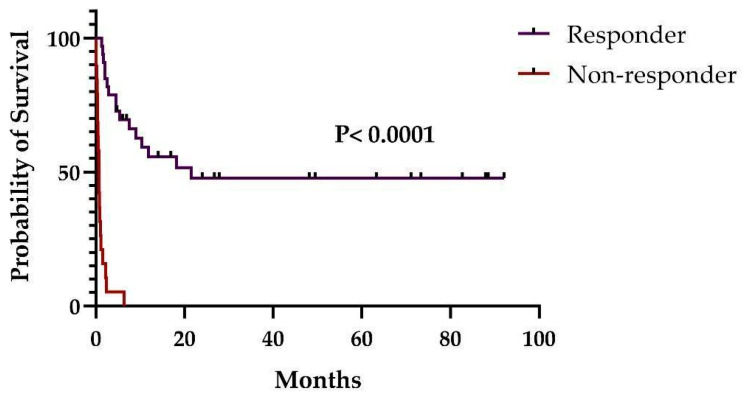
Overall survival according to response (complete and partial) at day 28 after MSCs.

**Table 1 pharmaceuticals-16-00512-t001:** Clinical characteristics of patients with steroid-refractory acute graft-versus-host disease.

	Total (*n* = 52)	Children (*n* = 35)	Adults (*n* = 17)	*p*-Value
**Age (years), median (range)**	12.5 (0.3–65)	8 (0.3–16)	32 (20–65)	
**Male (%)**	34 (65.4)	23 (65.7)	11 (64.7)	1.0
**Diagnosis (%)**				0.02
Acute leukemia/MDS	39 (75.0)	30 (85.7)	9 (52.9)	
Lymphoma	3 (5.8)	0	3 (17.6)	
Aplastic anemia	4 (7.7)	2 (5.7)	2 (11.8)	
Others	7 (11.5)	3 (8.6)	3 (17.6)	
**Conditioning regimen (%)**				0.002
MAC	41 (78.8)	32 (91.4)	9 (52.9)	
RIC	11 (21.2)	3 (8.6)	8 (47.1)	
**Graft source (%)**				<0.0001
BM	30 (57.7)	27 (77.1)	3 (17.6)	
PBSC	20 (38.5)	6 (17.2)	14 (82.4)	
UCB	2 (3.8)	2 (5.7)	0	
**aGVHD severity (%)**				0.37
II	2 (3.8)	2 (5.7)	0	
III	13 (25.0)	10 (28.6)	3 (17.6)	
IV	37 (71.2)	23 (65.7)	14 (82.4)	
**Organs affected (%)**				0.79
Skin	39 (75.0)	23 (65.7)	16 (94.1)	
Gut	50 (96.2)	33 (94.3)	17 (100)	
Liver	21 (38.5)	13 (37.1)	8 (47.1)	
**Days to MSCs infusion *, median (range)**	36.5 (4–294)	39 (4–250)	27 (12–294)	0.6
**N° of MSCs infusions, median (range)**	4 (1–11)	4 (1–11)	3 (1–8)	0.06
**MSCs Dose (×10^6^/kg), mean ± SD**	4.73 ± 1.30	4.93 ± 1.4	4.31 ± 0.98	0.11

* After aGVDH; MDS: myelodysplastic syndrome; MAC: myeloablative conditioning; RIC: reduced-intensity conditioning; BM: bone marrow; PBSC: peripheral blood stem cell; UCB: umbilical cord blood; aGVHD: acute graft-versus-host disease; MSC: mesenchymal stromal cells.

**Table 2 pharmaceuticals-16-00512-t002:** Overall response to MSC infusion according to the groups.

	Total (*n* = 52)	Children (*n* = 35)	Adults (*n* = 17)	*p*-Value
**OR day 28 (%)**	33 (63.5)	25 (71.5)	8 (47.1)	0.12
CR	19 (36.6)	17 (48.6)	2 (11.8)	0.03
PR	14 (26.9)	8 (22.9)	6 (35.3)	
**OR day 100 (%)**	19 (36.6)	16 (45.7)	3 (17.7)	0.06
CR	16 (30.8)	15 (42.8)	1 (5.9)	0.008
PR	3 (5.8)	1 (2.9)	2 (11.8)	
**OR day 180 (%)**	18 (34.6)	16 (45.7)	2 (11.8)	0.02
CR	17 (32.7)	15 (42.8)	2 (11.8)	0.05
PR	1 (1.9)	1 (2.9)	0	
**OR 1 year (%)**	15 (28.8)	13 (37.2)	2 (11.8)	0.1
CR	14 (26.9)	12 (34.3)	2 (11.8)	0.15
PR	1 (1.9)	1 (2.9)	0	
**OR 2 years (%)**	14 (26.9)	12 (34.3)	2 (11.8)	0.1
CR	14 (26.9)	12 (34.3)	2 (11.8)	
PR	0	0	0	

OR: overall response; CR: complete response; PR: partial response.

**Table 3 pharmaceuticals-16-00512-t003:** Adverse events associated with MSC infusion according to the groups.

	Total (*n* = 52)	Children (*n* = 35)	Adults (*n* = 17)	*p*-Value
**Adverse Events (%)**	17 (32.7)	12 (34.3)	5 (29.4)	1.0
Tachycardia/bradycardia	6 (11.5)	2	4	
Hypertension/Hypotension	4 (7.7)	2	2	
Dyspnea	2 (3.8)	1	1	
Nausea/vomit/abdominal cramps	7 (13.5)	7	0	
Headache/dizziness	5 (9.6)	4	1	
Seizures	1 (1.9)	1	0	
Fever/chills	5 (9.6)	4	1	
Rash/pruritus	2 (3.8)	1	1	
Angioedema	1 (1.9)	1	0	

**Table 4 pharmaceuticals-16-00512-t004:** Recent studies that evaluated the use of MSC in steroid-refractory acute graft-vs-host disease.

	Kebriaei (2020) [37]	Kurtzberg (2020) [29]	Kurtzberg (2020) [30]	Murata (2021) [38]	Ding (2023) [34]	Donadel ^c^ (2023)
**Type of study**	phase 3	phase 3	expanded access	retrospective	retrospective	retrospective
**Participants**	163	54	241	309	54	52
**Age ^a^**	43.8	7.0	9.6	49	12.5	12.5
**Severe aGVHD ^b^ (%)**	77.3	88.8	80.1	80.5	88.8	96.2
**Source of MSC**	BM	BM	BM	BM	UC	UC
**Lines of therapy**	≥3	2	≥2	≥1	≥3	≥3
**N^o^ of infusions ^a^**	8.8	9.9	11	8	2	4
**Dose per infusion (×10^6^/kg)**	2	2	2	2	2.54	4.73
**Time to infusion (days) ^a^**	-	12	23	29	19	36.5
**OR day 28 (%)**	58.3	70.4	65.1	56	59.3	63.5 (71.5) ^d^
**CR day 28 (%)**	-	29.6	14.1	24	44.4	36.6 (48.6) ^d^
**OS day 100 (%)**	-	74.1	66.9	-	-	51.9 (62.9) ^d^
**OS day 180 (%)**	34	68.5	-	40	-	38.5 (48.6) ^d^

aGVHD: acute graft-vs-host disease; MSC: mesenchymal stromal cell; OR: overall response; CR: complete response; OS: overall survival; BM: bone marrow; UC: umbilical cord; ^a^ Median or mean; ^b^ Grade III/IV or C/D; ^c^ present study; ^d^ (children).

## Data Availability

The data presented in this study are available upon request from the corresponding author. The data are not publicly available due to the privacy of participants.

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
