# Peer review of "Umbilical Cord Mesenchymal Stromal Cells for Steroid-Refractory Acute Graft-versus-Host Disease"

_pharmaceuticals, 2023, doi:10.3390/ph16040512_

Round 1

Reviewer 1 Report

Many studies suggested that MSCs were effective for SR aGVHD. MSCs are recommended as evidence level A-II for aGVHD treatment.[Lancet Haematol. 2020;7(2):e157–e167.] MSCs are multipotent progenitor cells that exist in various adult tissues, including bone marrow. This study evaluated the effect of MSCs from umbilical cord (UC-MSCs) for steroid-refractory acute graft-versus-host disease in 9 institutions, which is meaningful for clinical treatment. However, it is worth noting that age is an influencing factor for SR-aGVHD. This article divides cases into child and adult groups to analyze the efficacy of MSC, which interferes with the results.

Author Response

Point 1: Many studies suggested that MSCs were effective for SR aGVHD. MSCs are recommended as evidence level A-II for aGVHD treatment. [Lancet Haematol. 2020;7(2):e157–e167.] MSCs are multipotent progenitor cells that exist in various adult tissues, including bone marrow. This study evaluated the effect of MSCs from umbilical cord (UC-MSCs) for steroid-refractory acute graft-versus-host disease in 9 institutions, which is meaningful for clinical treatment. However, it is worth noting that age is an influencing factor for SR-aGVHD. This article divides cases into child and adult groups to analyze the efficacy of MSC, which interferes with the results.

Response 1: Excelent suggestion, which we included in this version in Introduction section: “Because of some encouraging results obtained so far, the European Society for Blood and Marrow Transplantation recommended MSC administration as second-line therapy (category 2A) for acute GVHD” [32].

Reviewer 2 Report

This is a well written article about the use of UC-MSCs for steroidrefractory acute GVHD in children and adults.

on line 84 delete ”great”

on line 86 replace ”remarkable ” with encouraging.

What was the viability of UC-MSCs at freezing and thawing respectively?Why was a dose od almost 5 times higher that generally used 1x10-6/kg?
Regarding safety add data from the multicenter studies by Lalu,MM et al Plos one 2012 and Thompson,M et al Clin.Med.2020.

Discuss also novel MSCs sources therapies for acute GVHD like the use of pooled bone-marrow derived MSCs from thee donors and placenta derived decidua stomal cells.

Author Response

Point 1:.on line 84 delete ”great”

Response 1: Done: “Subsequently, Kurtzberg et al, in two important studies, showed benefits of MSCs administration for children with aGVHD”.

Point 2:.on line 86 replace ”remarkable ” with encouraging.

Response 2: Done: “The first one is a one-arm phase 3 clinical trial that showed encouraging (overall response in 70.4% of the patients at day 28) clinical benefits for the patients treated with MSCs [29]”.

Point 3:.What was the viability of UC-MSCs at freezing and thawing respectively? Why was a dose of almost 5 times higher that generally used 1x10-6/kg?

Response 3: Viability just before freezing was superior to 95% and superior to 80% after thawing (included in the text-lines 130-131: “MSCs must meet the following criteria for clinical use: viability ≥ 70% (it was ≥ 95% at freezing and ≥ 80% after thawing);”.

The higher dose (the double of most studies, for instance, both by Kurtzberg), was decided before these most recent studies were published. Also, we thought that, as the dose at that time (planning) was not established (if it is now), we should administrate a high dose to control this severe complication, for which we had no other therapeutic option.

Point 4:.Regarding safety add data from the multicenter studies by Lalu,MM et al Plos one 2012 and Thompson,M et al Clin.Med.2020.

Response 4: References 42 and 43 included in the Discussion section.

Point 5: Discuss also novel MSCs sources therapies for acute GVHD like the use of pooled bone-marrow derived MSCs from the donors and placenta derived decidua stomal cells.

Response 5:  We included this information in the text with their respective references: “Other interesting sources seem to be pooled bone marrow and decidua MSC, also used to rescue aGVHD in children [45,46]. The decidua MSC was used in only six patients, with durable complete response in four of them”

Reviewer 3 Report

The article concerns a clinical trial of cellular therapy of severe steroid-resistant graft-versus-host disease (aGVHD) following hematopoietic stem cell transplantation using unrelated mesenchymal stem cells (MSCs) derived from umbilical cord blood. A total of 52 patients were treated at 9 transplant centers (a median of 4 cell infusions per case).  In general, the MSC therapy was associated with better overall and relapse-free survival. Some adverse effects were seen in 33% of cases, without severe side reactions. This therapy seems to work better in pediatric patients than in adults. 

Remarks

Materials and methods: Section 2.1.

- The study seems not to be randomized. It should be noted in discussion.

- Line 122- if mentioned, the cause of death within 24 h after MSC infusion should be explained.

Section 2.5. What scale was used for assessment of adverse effects in MSC therapy?

In general, the manuscript deserved publication since it provides a good evidence for efficiency of umbilical MSC therapy in steroid-resistant acute GVHD, thus being in agreement with numerous previously reported studies which used both allogeneic and autologous MSCs. Only minimal language editing is required.

Author Response

Point 1:. The study seems not to be randomized. It should be noted in discussion.

Response 1: Information included in Discussion section (first paragraph): “This non-randomized retrospective observational study showed that the administration of mesenchymal stromal cells obtained from umbilical cord results in substantial clinical benefits in steroid-refractory acute graft-versus-host disease, mainly in children”.

Point 2:.Line 122- if mentioned, the cause of death within 24 h after MSC infusion should be explained.

Response 2: Information included: “Two patients were excluded from the analysis, one because lack of data and another one because he died of sepsis before 24 hours after from MSC infusion (Figure 1)”.

Point 3:.Section 2.5. What scale was used for assessment of adverse effects in MSC therapy?

Response 3: Information included in the text (line 198): Common Terminology Criteria for Adverse Events (CTCAE).
